# Older People, Mobility and Transport in Low- and Middle-Income Countries: A Review of the Research

**Mark Gorman [1,\*], Sion Jones [1] and Jeffrey Turner [2]**

1   HelpAge International, London WC1H 9NA, UK; sjones@helpage.org
2   Gender, Inclusion & Vulnerable Groups, High Volume Transport Research Programme, IMC Worldwide Consultants, Redhill RH1 1LG, UK; jeffreymturner@hotmail.com
\*   Correspondence: mgorman@helpage.org

**Abstract:** Older populations are rising globally, which in high-income countries has helped to generate a growing literature on the impact of ageing on travel requirements and transport policy. This article aims to provide an initial assessment of the state of knowledge on the impact on transportation policy and usage of the increasing numbers of older people in low- and middle-income countries (LAMICs), through a review of the literature relating to older people and transportation. As both the academic and policy/practice-related literature specifically addressing ageing and transport in LAMICs is limited, the study looks beyond transportation to assess the state of knowledge regarding the ways in which older people's mobility is affected by issues, such as health, well-being, social (dis)engagement and gender. We find significant knowledge gaps, resulting in an evidence base to support the implementation of policy is lacking. Most research in low-income countries (LICs) is either broad quantitative analysis based on national survey data or small-scale qualitative studies. We conclude that, although study of the differing contexts of ageing in LAMICs as they relate to older people's mobilities and transport use has barely begun, institutions which both make and influence policymaking recognise the existence of significant knowledge gaps. This should provide the context in which research agendas can be established.

**Keywords:** ageing; disability; gender; mobility; older people; poverty; transport; urban

## 1. Introduction

There is by now a substantial body of literature discussing the impact of an ageing population in developed countries on travel needs and required changes to transport policy. An age-friendly built environment, including safe, affordable, and convenient transportation, has been identified as a critical factor in enhancing the quality of life for increasingly large numbers of older people. Studies across high-income countries have recognised that "Access to transport is regarded as a major determinant to achieving a good quality of life in older age" [1]. A number of reasons have been proposed to explain this growth of interest in ageing and mobility, the most immediate being the global increase in the absolute number and share of older people. This interest has also been facilitated by a substantial growth in the evidence available for high-income countries since 2000. However, very little is known about older people outside high-income countries [2]. This article aims to provide an initial assessment of the state of knowledge on the impact on transportation policy and usage of the increasing numbers of older people in low- and middle-income countries (LAMICs), through a review of the literature relating to older people and transportation. As both the academic and policy/practice-related literature specifically addressing ageing and transport in LAMICs is limited, the study looks beyond transportation to assess the state of knowledge regarding the ways in which older people's mobility is affected by issues such as health, well-being, social (dis)engagement and gender.

While there are many similarities between the experiences of mobility and transport use between high-income countries and LAMICs, there are also important differences. For example, transport infrastructure in many low-income settings is significantly poorer than for the "developed" world and some middle-income countries, and transport choices for older people who have to rely on public or shared transport services are extremely limited in many situations. It has thus been pointed out that context is significant, including, but not limited to, geographical settings: "Mobility is not just about the individual . . . but about the individual as embedded in, and interacting with, the household, family, community and larger society" [3].

This consideration of context also highlights the need for a review of research into age-related issues of mobility and transportation to move beyond a narrow biomedical model of ageing. For Schwanen and Paez context comprises four domains: the social relations of household, family, friends, acquaintances and community; the built environment, including transport, communication and other infrastructures; the institutions responsible for the built environment, policies and other forms of regulation; and the cultural norms and expectations that underpin travel practices and the regulation of mobility. In addition to these domains, life-course trajectories provide a history of past events and experiences for individuals, households, families, communities and societies, which impact significantly on issues to mobility in later life. So too do gender and ethnicity. Finally, old age, itself, is often treated as a fixed and stable category in transport studies [2]. This is notwithstanding that as long ago as the 1990s social gerontologists were pointing to the importance of understanding "the causal linkages between aging and social and political structures" rather than seeking inherent cultural meaning in the biological process of ageing [4]. Mobility and utilisation of transport is simply another example of how the meaning and definition of "old age" and "ageing" are culturally and geographically variable and socially constructed.

The article is structured as follows. After a review of the research materials and methods used we address global policy responses to ageing, mobility and transport, and this leads to a consideration of some of the evidence and policy gaps identified. This is followed by sections which address specific dimensions of ageing and its impact on mobility and transport. These are gender, social isolation and social support. Finally, consideration is given to ageing impacts and responses relating to mobility and transport requirements in humanitarian emergencies before a discussion of research gaps and emerging issues.

## 2. Materials and Methods

While this paper is not intended to be a systematic review, a literature search was undertaken utilising the Web of Science database, Google Scholar and other sources (including online material available from major international agencies). The search was undertaken between December 2018 and March 2019, and included material in English from 2000 onwards, just prior to the UN's first major plan of action on ageing relevant to the developing world in 2002. Criteria for inclusion was material focussed both on the experience of older people in utilising transport, and on the barriers due to age related to disability, gender and poverty. Search terms used were Older people AND mobility AND [Africa, Asia, Latin America]; Older people AND transport AND [Africa, Asia, Latin America]; Older people AND travel AND [Africa, Asia, Latin America]; Elderly AND mobility AND [Africa, Asia, Latin America]; Elderly AND transport AND [Africa, Asia, Latin America]; Elderly AND travel AND [Africa, Asia, Latin America]. The search on Web of Science database returned 82 results. After screening by title and abstract, 22 were selected for inclusion. In addition, a manual review was undertaken through Google Scholar and other sources which returned 69 relevant items from scientific and grey literature. Eighty-two items were identified through database searches and 22 items were included; 69 items were included from manual search out of a total of 81 items.

In addition to the Web of Science database, a manual search was also undertaken. The search terms selected here were chosen to extend the inquiry beyond a focus on ageing and older people (which would produce limited results) to include related issues such as gender, disability and poverty.

Countries selected aimed to achieve a representative geographical spread of LAMICs in Africa, Asia and Latin America. As articles and other publications were identified, citations they contained were used to identify other research of potential relevance on the World Bank's Open Knowledge Repository was searched, and the online resources of UN agencies (the World Health Organization, the UN Department of Economic and Social Affairs (which houses the UN's Ageing Programme), the UN Development Programme and UN Habitat. Other "grey" literature, included are the materials of Help Age International, and the publications of other non-governmental organisations working in related fields. (It should be noted that some of this material was developed for advocacy rather than objective research purposes and their conclusions should, thus, be treated with more caution). Our review also draws on the authors' extensive practitioner experience of work with older people, in a major international organisation dedicated to this field, including engagement with international and national policy makers, NGOs and communities in diverse LAMIC contexts. We are confident that we have been able to adequately identify eligibility and quality of the material we present in this paper.

## 3. Results

### 3.1. Ageing, Transport and Global Policy

The broadest context for population ageing is at the global level. Demographic change and population ageing are now global trends, not ones confined to high-income countries. The growing numbers and proportions of people living into later life in all societies pose new questions about their transport and mobility requirements and the extent to which transportation policy and practice will be responsive to the mobility needs of older people. The policy response at the global level is an indicator of the extent to which demographic ageing has elicited a policy response, and the degree to which an evidence base to inform policy has been developed.

At a global level, policy parameters are now set by the Sustainable Development Goals (SDGs), whose call to leave no one behind requires that the SDGs are met for all of society, at all ages, with a particular focus on the most vulnerable, including older women and men [5]. Even before the advent of the SDGs various UN agencies addressed the impact of transportation issues on ageing populations in the context of wider agenda-setting debates. Prior to the United Nations' second global assembly on ageing in 2002, Kalache and Keller noted the importance of adequate public transport to enable more independent life well into very old age, noting that "one of the major challenges is to ensure access...to all older persons—including the poor and those who live in remote areas" [6]. The UN's international plan of action on ageing which emerged from the 2002 World Assembly included as a priority "Ensuring enabling and supportive environments—Transportation is problematic in rural areas because older persons rely more on public transport as they age and it is often inadequate in rural areas" and called for investment in local infrastructure, such as transportation [7].

The UN's New Urban Agenda (which lays claim to being "a paradigm shift based on the science of cities; it lays out standards and principles for the planning, construction, development, management, and improvement of urban areas") also makes broad commitments to be age-inclusive, from data collection through consultation to policy-making. Specifically in relation to urban transportation, commitments are made to enable access to safe, efficient and sustainable transport systems for all [8].

In its Global Ageing and Health Report (2015) the World Health Organization addresses the interactions of the different dimensions of an older person's context (social relations, the built environment, policy and regulation, and cultural norms). "Mobility is influenced not only by an older person's intrinsic capacity and the environments they inhabit but also by the choices they make. Decisions about mobility are, in turn, shaped by the built environment, the attitudes of the older person and of others, and having both a motivation and the means to be mobile (such as by using assistive devices or transportation)". The Report goes on, stating: "Specific consideration will need to be given to the needs of older people to ensure that environments are accessible, including homes, public spaces and buildings, workplaces and transportation" [9].

The World Bank has also recognised that "Vulnerable and special-need groups (including women, children, persons with disabilities, and older persons) are underserved by public and private transport systems...because users and providers do not carry the full societal costs of excluding vulnerable groups". The World Bank argues for "equity and inclusivity [to be] at the heart of Universal Access. This objective...places a minimum value on everyone's travel needs, providing all, including the vulnerable, women, young, old, and disabled, in both urban and rural areas, with at least some basic level of access through transport services and leaving 'no one behind'" [10]. Both the Sustainable Development Goals and the New Urban Agenda call for expended, age- and gender-responsive public transportation that responded to the challenges faced by all.

More broadly, the NUA calls for age-and gender-responsive approaches to policy and planning processes, design, budgeting, implementation, evaluation and review, and this indicates the important potential of focussing on universal access, rather than transport-related solutions aimed at older people alone. The World Bank has addressed ways to ensure universal access, but again the research basis is lacking, at least as far as the participation of older people is concerned. The World Bank itself notes that "While there is no widely agreed upon method of measuring universal access, there is a general agreement that sustainable transport should leave no one behind. Data that measure access to transport infrastructure and services for urban areas are not readily available on a global scale. The data that do exist suggest that the accessibility gap is huge, and potentially growing . . . " [11].

### 3.2. Evidence and Policy Gaps

Notwithstanding the concern of the UN and other global bodies to establish these broad declaratory frameworks, evidence-building and analysis have not been prioritised. For example, while the World Bank makes significant reference to the evolving requirement for sustainable and accessible transport arising in part from the changing requirements of older populations, no supporting evidence is cited for statements such as "The ageing of the population is likely to have significant effects on mobility..." The evidence base for a research agenda to support the implementation of global policy is therefore lacking [11]. Moreover, these global policy concerns are very rarely reflected at regional and national levels. National policymakers cite resource constraints and the absence of evidence for these policy gaps [12]. Thus, while a number of studies conclude with proposals for policy interventions, these have not been translated into practice [13,14].

More recently greater attention has been paid to the global implications of population ageing for transportation and transport policy [15]. A number of studies in LAMICs have been undertaken, often with the stated intention of influencing transport policies. A study of Hong Kong, for example, took as a starting point the rapid growth of the older population, which was expected to significantly affect the public transport systems, and "to provide suitable policy recommendations that cater to the travel needs of an ageing society" [16]. Other work has relied on the WHO SAGE data for China, India, Mexico, Ghana, Russia and South Africa, which are, thus, studies of small samples of older populations in middle- rather than low-income countries [17]. Datasets comparable to the SAGE data do not exist for low-income countries [18].

Other countrywide studies are rare and those that exist use broad national datasets, such as USAID's Demographic and Health Survey. Although the DHS is conducted in around 90 LAMICs and provides data for approximately 30 indicators in the Sustainable Development Goals, its data on older people are of limited value. While interviewees are women aged 15-49 and men aged 15–49, 15–54, or 15—59, data on older people are collected only through a whole-household questionnaire. Only two countries (South Africa DHS 2016, Haiti DHS 2016) have extended or lifted the upper age cap in individual questionnaires. Thus, in sub-Saharan Africa for example, only one significant national transport study has been undertaken with older people [19]. As we have seen, urban studies are confined to West Africa, particularly Nigeria. For rural areas the literature is very meagre [20]. As we have noted, studies in both south Asia and Latin America also concentrate heavily on urban settings.

These significant research gaps should be seen in the context of the overall lack of both empirical data and theoretical development on ageing and older people in LAMICs [14]. This in turn impacts on planning and policy formation. Commenting on the lack of data sources for an inquiry on active ageing in the Philippines, for example, Pettersson and Schmöcker note that better datasets are a basic requirement for urban planning in developing and newly developed countries [21]. Constraints such as these apply particularly in Sub-Saharan Africa where, despite an intensification of debate on ageing and formal expressions of commitment on the part of national governments, comprehensive policy action is still lacking. "The impasse reflects a lack of political will and an uncertainty about required policy approaches, engendered by wide gaps in understanding...in the region" [21]. A review of research relating to care-seeking behaviours among older people finds that apart from a number of small-scale, qualitative studies, no systematic, country-level evidence exists. Their findings, which point to the negative impacts of physical and logistical access difficulties and financial barriers related to service fees and/or transport costs, thus have no means of translation into policy-making [22].

This highlights a particular problem in relation to population ageing and older people. Parker et al. point out that "Aging in the west is often viewed from a biomedical perspective where the emphasis is on medical treatment and health and social care arrangements. Biomedicine also dominates international health strategies, organizations, and the funding streams for aid..." [12]. From this perspective, the wider implications of ageing as another stage in the life course receive much more scant consideration, and the familial, social and economic relations which structure older people's lives tend to disappear from view. Furthermore, older people are often excluded from data collection mechanisms; for example, much of the data available for LAMICs comes from household surveys, which set age restrictions and, therefore, do not routinely include older respondents. They often provide data and analysis at the household rather than the individual level, and data, where collected, is not disaggregated and analysed by age [23]. Exceptions to this, such as the World Health Organization's "Study on global Ageing and adult health" (a longitudinal study collecting data on adults aged 50 years and older), have very small sample sizes and tend to focus on middle-income countries where datasets may be more reliable than in the least-resourced settings [24].

The inattention to the needs of older populations in transport policy may also reflect the relative lack of political participation by economically and socially disadvantaged older people, and their consequent inability to influence decision-making on mobility services and investment. A low level of political participation has been noted as a key measure of the social exclusion of older people in middle- and low-income countries [25]. While the participation of older people in decision-making processes is restricted, the relative weight given to perceived economic value, social participation and reducing inequalities is likely to remain limited. Porter et al. point out that "Transforming evidence into policy and practice is particularly challenging in the transport sector which is dominated by male, middle-aged, middle-class engineers whose principal focus is road construction rather than transport services and where there is still a common reluctance to engage with users or with qualitative data" [20].

The lack of evidence that the perspectives of older people, amongst others, are considered when planning public transportation has been noted by, for example, the Institute for Transportation and Development Policy (ITDP), which recognises the democratic deficit created as result of local and national government transportation investment and planning decisions which fail to consult with and include urban residents, particularly those who are often the most marginalised [26]. In terms of investment and impact assessment, even in high income contexts, there is little existing guidance for comprehensive transport equity analysis that includes all groups of people [27].

*3.3. Ageing, Health and Mobility*

The relationship between ageing and health is complex; it is now well recognised that a global epidemiological transition from diseases, which mainly impact children, to non-communicable diseases (NCDs), which are more common in adults, has accompanied the demographic transition

with population ageing at its core. The combination of disability and ageing potentially provides a significant limitation on the mobility of older people in LAMICs, though again the evidence base is relatively sparse, since very little research has been conducted on moderate to severe disabilities affecting mobility, communication and mental function in later life [28–30]. For example, a comparative study across urban settings in Latin America and the Caribbean, while addressing gender differences in later life health and functional status, similarly did not establish a connection between limited functional capacity and transport utilisation [31], this notwithstanding that the research utilised data from the SABE study ("Survey on Health, Well-Being, and Aging in Latin America and the Caribbean"), the first health study of the old people in Latin America and the Caribbean to include transportation among the physical environment determinants which it assessed [32].

What does seem to be clear from the evidence is that the outcomes of disease and injuries are increasingly undermining the ability of the world's population to live in full health. A recent review of data from the WHO "Global Burden of Diseases, Injuries, and Risk Factors Study" noted that (after anaemia) the leading cause of impairments (by number of individuals affected) were hearing loss and vision loss. These sensory organ disorders were also the leading causes of impairments in 22 countries in Asia and Africa and one in Central Latin America, while lower back and neck pain was the main cause of disability in most countries [33]. There is some recognition that these disabling conditions can be critical for the wider social participation of older people. There is little published data on the potential health benefits of active travel in low and middle-income countries, although some evidence exists. For example, studies (which again drew on WHO's widely-used SAGE data for six MICs, or the related INDEPTH datasets) found a correlation between increasing age and reduced active travel (walking or cycling), translating into a higher risk of being overweight and raised BMI [34,35]. A smaller-scale study in peri-urban areas of Nepal elicited similar results [36].

Problems over access to health care facilities is the most frequently cited transport-related issue for older people in LAMICs, with physically remote clinics and hospitals necessitating costly and difficult travel a key barrier. A common finding was that poverty and mobility constraints combined to reduced older people's access to healthcare [37]. Data from the WHO's World Health Survey (2002–2004) indicated that, in total, more than 60% of older people in LAMICs did not access health care either because of the cost of the visit, or because they did not have transportation, or they could not pay for transportation. Transportation may be a particularly important issue for older people who live in rural areas because services are often concentrated in large cities far from people's homes and communities [38]. A study in rural Kyrgyzstan, for example, found that one in five older people lived more than a 30-minute travel away from a health facility, with access particularly problematic for those with a limiting longstanding illness or disability [39]. The structure of health care provision is also problematic, noted by a study in South Africa of care and treatment for older people (50+) living with both HIV and other chronic conditions, which found that services were typically provided at different health facilities or by different health providers, necessitating multiple patient journeys [40].

Psychological factors, such as depression, may also play a role in limiting mobility. A study to identify the most burdensome functioning domains in depression and their differential impact on the quality of life using SAGE data from countries in Asia, Africa and Latin America found that affect, domestic life and work and interpersonal activities were the domains most affected by depression, with gender also playing an important differentiating role [41]. A lack of self-confidence in physical capacity, leading for example to a consideration of the risk of falling, as well as concerns regarding traffic, have also been identified as limiting factors. The risk of falls is a prevalent factor in activity restriction by older people in a variety of settings. Even among a physically active older population in the Colombian Andes the risk of falling decreased physical activity with negative effects for self-perceived health and depressive symptoms [42–44].

Again, however, and notwithstanding the high and rising level of death and injuries from road traffic accidents in LAMICs [45], and the significant proportion of older victims in some contexts, research on the impact of traffic accidents on older people remains undeveloped [46]. This may reflect

a context in many low- and middle-income countries, where poor quality transport infrastructure (including poorly maintained roads, pavements, crossings, shelters, as well as poorly maintained and inaccessible vehicles) affect people throughout the life course. Accident rates are high, affecting whole populations. In these circumstances it is perhaps unsurprising that the specific impacts on older people receive relatively little attention.

The evidence that does exist indicates that older people, whether walking or using public transport, have a significant exposure to accidents and injury [47]. Even crossing roads in busy urban environments in LAMICs may (rightly) induce feelings of anxiety and fear, with a consequent impact on quality of life [48]. Although data limitations mean that studies have tended to be small-scale, they nevertheless raise important issues. For example, a study of the constraints on the travel of older people in a Nigerian city found that issues such as vehicle design, long access and waiting time as well as poor facilities at bus terminals were serious constraints to the effective mobility of older people [49]. Similar conclusions were drawn from a study in Pakistan, which pointed to safety and security issues, as well as attitudinal behaviours on the part of service providers, as key factors in the mobility and transport utilisation of older people [14]. These are issues to which we return below.

### 3.4. Ageing, Transport and Gender

As noted above, a key cross-cutting issue affecting the mobility of older people is that of gender, which has now become a well-recognised issue in transport research, and one that intersects with other factors to increase disadvantage. A study in Bogota, for example, which summarized recent research on unequal access to transport systems, focused on the ways that gender and socioeconomic inequalities may be exacerbated by differences in transport accessibility [50]. Nevertheless, again the intersection of gender with age has attracted far less attention, despite this being a feature that research has shown to be clearly recognised by older people themselves [51]. Moreover, there are no systematic gender and age inclusion procedures for transport, either in terms of training of professionals, participation of users or the design and planning of systems, services and equipment. Again, as international institutions such as the World Bank have pointed out, a lack of evidence limits progress in policymaking, particularly regarding gender issues in transport relating to older people in the developing world [52].

While other international institutions have drawn attention to the intersections of age and gender, again, a clear evidence base substantiating their assertions is lacking. The UN Economic and Social Council (ECOSOC) for example, has pointed out the economic impacts of differential access to transport for women. They noted in 2009 that women in low-income countries were seriously constrained in their access to transport, limiting access to labour markets, increasing production costs and reducing the amount of goods which could be taken to market. The ECOSOC report focussed on issues such as poor access to transport, affecting girls' school attendance, women's use of health and other public services and maternal mortality. While older women were not specifically mentioned, evidence was noted regarding the lack of access to transport services affecting women who spend long hours hauling water and fuel and walking to and from farm plots. Head-loading was cited as a major health hazard to women, as was the potential to suffer higher accident rates through walking on crowded roads with heavy burdens [53]. Other evidence indicates that these are impacts which fall on women who continue to work into later life [20].

A small number of studies do address age and gender as features affecting access to transport and impacting on mobility. Research in urban settings in Pakistan, Iran and Malaysia all found that age and gender, together with other factors, such as car ownership, travel time, travel cost, household size and income, were significant factors in influencing individual choices in transportation. These studies all included higher-income households, and demonstrated the importance of car ownership in travel frequency, though even here gender played an important role, with older men making significantly more journeys than older women [14,54,55]. Other evidence points to the limitations in "choice" that older women in particular may have; a study of a urban setting in Nigeria found that gender, along

with increasing age, education and monthly income, were significant in determining walking as the mode choice, in a situation where over 70% of older people lacked access to motorised transport [56].

The impact of gender on transport access in Africa is also examined in a recent field study which assessed whether gender mainstreaming in rural transport programmes in Tanzania has had a transformative effect on women. The study found that despite the attention given to gender issues, women's participation in designing and implementing rural travel and transport programmes was limited by negative views of women's potential to contribute effectively to such programmes. On the other hand, road construction did lead to improvements in transport services and expanded travel options for all women, including those in later life, who had more time both for family and to pursue multiple projects [57].

### 3.5. Mobility, Transport and Social Isolation

Gender thus has clear impacts on mobility and helps to bring into question the role that mobility plays in enhancing or mitigating loneliness and social isolation. The World Health Organization has recognised the importance of attitudinal factors for older people, commenting that "Mobility is influenced not only by an older person's intrinsic capacity and the environments they inhabit but also by the choices they make. Decisions about mobility are, in turn, shaped by the built environment, the attitudes of the older person and of others" [9]. Characteristics of the built environment can function to restrict older people's mobility and participation in urban life [58,59]. Nevertheless, research evidence for the role that transport access plays in influencing older people's social isolation is sparse for low-income countries.

In recent years a significant number of studies have made a direct causal link between transport accessibility and social exclusion [60]. Again, the great majority of these studies have been undertaken in high-income countries where both income poverty and lack of transport are relative rather than absolute states. There are few studies on the relationship between transport and social disadvantage in LAMICs, where income poverty is absolute and where access to transportation is very limited [61]. Studies addressing the social exclusion of older people in relation to transport in low-income countries are still rare [62–64]. While some studies discuss the part played by physical mobility in older people's social isolation, the role that transport access plays in this is not examined [65].

For those living in LICs, social isolation is a significant risk with increasing age, again mediated by factors such as gender and poverty. A primary issue for older people in these situations is the impact of psychological factors. WHO notes the importance of the attitudes of older individuals and of others in decisions about mobility, and the motivation (and the means) to be mobile [9], p.180. A small number of studies has addressed the links between confidence and behaviour, to assess the influences on older people's decision-making relating to mobility. Fear of crime and concerns over the safety of public transport have been identified as a limiting factor in older people's mobility in a number of different settings, albeit with wide national and regional variations [66].

WHO studies highlight a number of issues that influence decision making related to older people's use of transportation. These include availability of services, affordability, reliability and frequency, appropriateness of service destinations and the availability of specialised and priority services. Comfort is also highlighted as a key concern, with respondents to focus groups in studies in Rio de Janeiro and Mexico City citing hard, uncomfortable and bumpy journeys exacerbating existing health concerns and discomfort. Similar concerns were raised by focus groups undertaken in five cities in Argentina, with issues such as struggling to board buses because of the height of the initial step. Multiple elements are necessary to make public transport an attractive alternative for older people, including physical accessibility, the availability of information and ease of way finding [67].

Social exclusion has a number of features which may be identified with the experience of older people in relation to transport in the developing world, and which have been the subject of studies which address older people's mobility and transport needs only obliquely. These features of social exclusion are broader than income poverty alone, and include a lack of participation in social, economic

and political life. They are also multidimensional and cumulative: for example, limited financial resources and security are often reciprocally linked to low education and skills, ill-health, and, as noted above, limited or no access to political influence. Social exclusion is also dynamic, and subject to changes over time, as well as directly affecting individuals and households as well as neighbourhoods and local communities [68]. Curl and Musselwhite have pointed out that, despite policy and discourse (at least in high income countries) placing strong emphasis on the maintenance and extension of independence and "ageing in place" as vital requirements for a dignified healthy later life, changes in later life income, lifestyle and ability to use transport present significant challenges for the provision of appropriate transport services [15]. Where and how people live impacts day-to-day life, particularly in the urban built environment. Spatial barriers interact in complex and specific ways with the intersecting identities that individuals carry, creating unique patterns of disadvantage. For example, lower income people (in which older people, together with other disadvantaged groups such as women, children, are also disproportionately represented) are increasingly pushed to the periphery of cities in their search for affordable housing [26].

In some cases, this is as a result of deliberate government policy and this has an impact on access to affordable transport. This was noted by a study of peri-urban areas in South Africa, which found that location along the urban-rural continuum significantly affects both transport expenditure levels and the perceived severity of transport affordability problems for marginalised people, notably those with disabilities and older people [19].

### 3.6. Mobility, Transport and Social Support

One important motivation for mobility and travel for older people is to access social support, particularly from family members, and a factor mitigating against the need for mobility in many low-income settings in Asia remains the fact that normative support for filial obligations to ageing parents is widespread. A recent study of older people in Myanmar for example, found that the majority of rural-dwelling older people had an adult child co-residing or living nearby, facilitating intergenerational exchanges of material support and personal services, and reducing the need for travel [69]. In the same vein, a review of studies on long-term care systems in sub-Saharan Africa noted that "the provision of long-term care rests overwhelmingly with family members, in line with customary sub-Saharan African norms of family solidarity and obligation". However, the review concluded that the evidence also revealed that "a substantial group of older people received no family care whatsoever" [70]. Again, studies which address the implications the care received by older people in terms of their mobility and transport requirements are almost non-existent for low-income countries. This is equally the case for evidence of older people as caregivers, despite the recognition of the important role that they play in many contexts. A study in Kenya, which found that older women AIDS-caregivers reported high disability scores for mobility and low scores in self-care and life activities domains, indicates that this comes at a high cost to older individuals [71]. Similarly, a Ugandan study showed how older women caregivers faced drastic disruptions of living arrangements, including lengthy travel times and absences from their homes to care for PLWAs [72]. Nor should it be assumed that care-giving is ensured simply by the traditional norms of extended family relationships, when families are spatially separated. For example, a recent study of the mobility constraints experienced by married and externally-resident daughters providing end of life care to parents in northern Ghana shows how these younger women had to negotiate conflicting responsibilities to provide parental care [73]. Similar issues are identified for settings in a recent comparative study on Manila and London [74].

Urban locations are by no means the only settings where security issues play a part in older people's travel choices. For example, both the lack of paved main roads in rural Myanmar and security issues in some regions have been found to be a significant barrier to older people's access to health services [69]. A study in Papua New Guinea examining the impact of road development on people with disabilities found that while road development improved service access, inaccessible road and

transport infrastructure remained insurmountable barriers to easy and safe travel. Roads were planned for the needs of vehicle users, and planning around road infrastructure did not involve consultation with people with disabilities [75].

*3.7. Mobility and Transport in Crises and Humanitarian Emergencies*

The particular vulnerabilities of older people in times of systems breakdown is an area which has had some attention in relation to humanitarian response. Clearly mobility plays a decisive role in periods of crisis and humanitarian emergency, and is a major potential issue for older people, (as it is for people with disabilities and children). This has elicited some response from humanitarian agencies which have developed guidance on making services accessible to older people with mobility constraints [76]. However, in comparison to the attention paid to people with disabilities, older people have received less attention from the international agencies, and their transport needs have been largely ignored. The United Nations High Commission for Refugees (UNHCR) has produced guidelines on "Working with Older Persons in Forced Displacement", but while the companion guidance on working with people with disabilities in forced displacement has recommendations for accessible transport, that for older persons does not, confining itself to a broad recommendation to "Help older persons to access services by providing transport" [77].

More recently, a consortium of humanitarian agencies and academic institutions working in the fields of disability and ageing have recognised the overlapping and often coterminous vulnerabilities of older people and those with disabilities in emergencies in establishing key standards for achieving the inclusion of both groups in humanitarian action, by, for example, addressing barriers that affect participation and access to services [78]. However, a more specific focus on transport requirements remains lacking, reflecting a lack of studies on the specific mobility and transport requirements of older people in emergency situations.

The work that has been done reveals some familiar issues. For example, a recent study, based on focus group interviews with older displaced persons in Sudan, found that many older people with disabilities faced a number of physical barriers such as having to travel long distances to distribution points, a lack of accessible transport, as well as inaccessible housing, toilets and public buildings. Family and friends were identified as key providers of both physical and financial assistance, including, notably, paying transport costs, but the cost of transport to key points, such as health facilities, was a constant source of stress [79].

## 4. Discussion

This review indicates that, notwithstanding growing recognition (at least at a global institutional level) of the implications of population ageing for transport policymaking, this has yet to be translated into significant investment in research in LAMICs. Such research as has been done in LICs is small-scale and based on qualitative evidence, which may be dismissed by policymakers as "anecdotal". This lack of systematic national-level evidence makes translation of any research findings into policy highly problematic. We have noted the predominant role that transport infrastructure planning plays in many LAMICs, to the detriment of consideration of social value. Such policy biases reflect the fact that policymaking is not a rational, evidence-based process, but is the outcome of numerous interactions between policymakers and other actors (including researchers, but also politicians, lobby groups and advocacy organisations). While evidence plays a role in illuminating policymaking, so too do other factors, such as political voice. As we have seen, the relative lack of political participation by disadvantaged older people limits their ability to influence policy and, by extension, research decisions. This may be both a cause and effect of the exclusion which we have noted of older people from many data collection mechanisms.

There is equally limited consideration of the ways in which age intersects with other factors, particularly gender, differential physical and mental capacity, and poverty. Here we have seen that the lack of age-disaggregated data is problematic. The imposition of upper age limits in data collection

is a significant barrier to understanding the characteristics of ageing, whether at the individual or the population level. Furthermore, despite the diversity to be found within older populations, disaggregation by age has been analysed very little. The lack of gender disaggregated data and analysis is also noticeable, notwithstanding the frequent references in policy documents to the particular challenges of public transport for older women.

Access to services is often a strong focus of current work regarding older people's mobility. The difficulty of accessing health services is, as we have seen, frequently cited, but there is a limited perspective on the wider needs of older people beyond basic health. Issues of mental health, social isolation and loneliness are rarely discussed. The roles played by older people as care-givers or recipients, and their mobility implications has become an increasingly significant area of research in Western contexts, and is also an important issue in LICs. This is an area that clearly needs greater attention. Research on mobilities across generations is also a major gap, with little attention paid to relational mobilities, despite the clear importance of intergenerational connections, notably (but not exclusively) related to care-giving.

However, we have also noted an emphasis in research related to population ageing which focusses attention on medical and social care arrangements, to the detriment of considering the social and economic contexts in which people live the later stages of their life course. We have noted that some work has been done in this regard. For example, a number of studies have emphasised the important place that the continuing need to earn a livelihood has for many who enter later life in poverty. Older people's mobilities and the role of transport in relation to livelihoods is, thus, another important area of inquiry but, again, is a significant research gap in LICs (where income earning must continue into old age in the absence of adequate pension provision). In this regard, the affordability of transport, economic needs, subsidy issues and income categories are all areas for research which require further attention [19].

Research undertaken on psychological effects which act to limit the mobility of older people, and the impacts of social isolation, has been reviewed. Issues ranging from concerns over road safety to harassment, personal security, stigma, shame, discrimination and the impact of crime have all been shown to pose significant barriers to older people's mobility. Many transport services, whether public or private, provide physical hazards for older people, as does poorly maintained physical infrastructure. Again, with limited exceptions, these problems are inadequately addressed in the research literature. While the potential value of virtual connectivity (through mobile phones for example) to replace or complement physical mobilities has been examined, both the benefits and costs of virtual connectivity remain to be researched further [20].

In the absence of access to relevant national-level quantitative data, it may be useful to consider other innovative research approaches. For example, action research, involving interventions followed by in-depth monitoring of impacts over a period of time, which involve older people as active research participants, has potential. However, while there is a growing rhetoric around the involvement of older people in research process, and some action has been taken, older people tend to remain respondents. The value of using co-investigation approaches has been demonstrated by work in Tanzania and in Papua New Guinea [19,20]. New methodologies, such as the use of geo-mapping alongside participatory inquiry methodologies to explore the social and spatial barriers to access of urban services, also need further review and analysis [20]. To develop the necessary expertise, in academic institutions, governments and NGOs, to conduct mobility studies with older people there is also a need to build research capacity, both in-country and external expertise.

This review has indicated the importance of taking account of great diversity in the ageing experience, across widely varying contexts. Influences on ageing range from societal and political attitudes to older people to the built environment, population density, climate, topography and land use. As we have aimed to show, study of these differing contexts of ageing in LAMICs, particularly as they relate to older people's mobilities and use of transport, has barely begun. At the same time, as we have seen, policymaking institutions recognise these issues, and assert the importance of prioritising

the widest possible inclusion in policies promoting transport and mobilities of people at all ages, so that, in the words of the Sustainable Development Goals, "no one will be left behind" [80]. We have seen that those institutions which both make and influence policymaking recognise the existence of significant knowledge gaps, some of which have been discussed in this article. This should provide the positive context in which research agendas to answer some of these key questions can be established.

**Author Contributions:** The article was conceptualized by M.G., S.J. and J.T.; the methodology was developed by M.G., who also undertook the validation, and formal analysis.; the investigation was undertaken by M.G. and S.J.; the original draft preparation was by M.G.; the review and editing was by M.G., S.J. and J.T.; project supervision was by M.G and J.T. and project administration was undertaken by M.G.

**Funding:** This research was funded by the UK Department for International Development.

**Acknowledgments:** This research was funded by UK AID through the UK Department for International Development under the High Volume Transport Applied Research Programme, managed by IMC Worldwide. Special thanks to Gina Porter for her support in developing the literature search and reviewing the output.

**Conflicts of Interest:** The authors declare no conflict of interest.

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
