# Peer review of "Older People, Mobility and Transport in Low- and Middle-Income Countries: A Review of the Research"

_sustainability, doi:10.3390/su11216157_

Round 1

Reviewer 1 Report

The review explored in this article can be of substantial scientific and clinical interest to the understanding of the role of mobility and transports for decreasing of disability and enhancing of functioning of older adults. Thank you very much for the opportunity to review the manuscript.

Introduction and the rest of the text is a narrative without critical analysis.

Overall the manuscript is clear, even so a review of the evidence needs a clear description of the methods. I will outline below my main concerns:

Study design, Literature search strategy (who performed the search, databases, key search terms, eligibility criteria, ...), Study selection, Quality assessment of studies, Studies analysis are some of the topics that are missing in the Methods section.

Discussion section will be important to separate from results.

Keywords need corrections (for example, “transport” is not included);

In conclusion, the paper has good potential after authors clarify those pointed limitations.

Author Response

Firstly, many thanks to the reviewers for their careful consideration ofthe article and for their helpful comments.

In response to reviewer 1's comments I have revised the article as follows:-

A methods section has been included, responding to the comments made, on the search strategy and quality assessment. The study was not a systematic review, but was grounded in the authors'e practitioner experience in the field of ageing in Low- and Middle-Income Countries. The study thus drew on personal knwowledge, which informed a careful study of both published and grey literature relevant to the the theme. A cross-section of countries across Asia, the Middle East, Africa and Latin America and the Caribbean was selected, and a keyword search list was drawn up which extenede beyond ageing and older people to broaden the focus somewhat by including (for example) disability. Google scholar and web of science databases were utilised. It is acknowledged that the quality of gtrey literature is more difficult to assess, and this is acknowledged in the article.

The discussion section has now been seperated from the results. Those references which remain in the discussion section are I believe relevant the conslusions under disscussion. 

The keywords have been revised.

Reviewer 2 Report

This is a useful and interesting review of a subject of rapidly rising importance that has been woefully neglected: the suitability of transport infrastructure for older people. The article provides a review of the literature available on older people’s mobility in low and middle income countries from the perspective of research that could inform policy development.  The underlying perspective is that old age is not a fixed and stable category but that ageing, frailty, and later life are the expression of social and political structures that shape the outcome of life course trajectories by gender, ethnicity and class (here framed as poverty and marginality).

The strength of the article lies in the critical analysis of data sources and research methods and the difficulty of establishing the evidence that policy makers will not dismiss as ‘anecdotal’ while not falling back into the trap of a biomedical model of ageing which, in some forumulations, can strip out societal and distributional effects on later life.

This critical analysis of the intersection of data sources, research methods and policy agenda formation/implementation, and the implied view that infrastructural provision should be as much about people and their rights as about the economy, gives the paper relevance beyond mobility in later life – yet another strength.

I would suggest some changes to reflect the strengths of the article.

change the title to Review of the Research, as opposed to Review of Evidence.

the organization of the paper needs a little work, these are all minor but changes that will bring the paper’s strengths into clearer focus:

i. the introduction needs shortening with a clearer setting out of the argument and how the paper will be structured. Some paragraphs on research on transport would be more appropriate later in the paper.

ii.  the introduction needs a paragraph on the underlying methods for the paper, now currently only in the abstract.

iii. it would strengthen the paper if lines 168 to 195 could be turned into a separate section discussing policy gaps, the lack of funding for research, the differential regard for and misunderstanding of what different research methods can bring and policy/research biases in transportation investment and planning.  A critical discussion of Ln 45-9, that apart from a number of small-scale qualitative studies no systematic country-level evidence exists which means that their findings have no means of translation into policy making would be especially valuable.  For policy makers what counts as systematic country-level evidence? This is of particular interest as in a number of places in the article there is an implicit suggestion that the WHO SAGE studies could be viewed as systematic country-level evidence, yet their sample size is miniscule by comparison to their population size. Wave 1 surveyed 34,124 people aged over 50 across 6 countries with a great deal of in country diversity and of which two alone, China and India, had more than 2 billion people between them (Kowal 2012).

iv. the subhead ageing, health and mobility could do with a short discussion of road use and road safety, including the extremely high death and accident rates in many countries of the south for the general population let alone older population. This would bring into focus, early in the paper, the recognition that transport is not just the means for transporting people but the conduits of travel – including roads, crossings, shade, paving, shelters etc.  This reorganization would entail breaking up section 6, on psychology and mobility, which, following the reviewed research, takes at face value the characterizing of older people’s rational consideration of risks (and the significance of the jeopardy involved) as ‘fear’.  Security issues, including their lack, could go into the earlier section on social support, providing a further example that widens the concept from family to community and society.

v. the methods section should be broken up and integrated into the earlier parts of the paper, including the conclusion. Lines 474 to 478 are intriguing and more information on the limitations of the Nigerian example of quantitative surveys which were not based on prior qualitative research would be a useful case study for use earlier in the paper.

Overall a paper that brings together important questions about the relationship between policy agendas, the differential weight accorded in policy circles to different research styles (including whose voice counts) and the outcome for older people’s (and others’) mobility.  I look forward to reading the final version.

Author Response

Firstly many thanks to the reviewer for his/her careful consideration of the article and helpful comments. In response I have revised the article as follows:-

The article's title has been changed to "review of research",

The paper has been reorganised as suggested. Specifically:-

The introduction has been shortened, with a setting out of the argument pursued in the article and a paragraph on the structure of the paper. Paragraphs on research of transport issues have been removed from the introduction and included later in the article.

A methods paragraph has been included. 

Lines 168-95 have been restructured and added to in a new section on policy gaps and biases. I have attempted to respond to the issue of bias over different research methods and the extent to which they are taken into account in policymaking. Regarding the point as to what counts as systematic evidence I acknowledge that there may be an implicit suggestion that SAGE studies are acceptable as systematic country-level evidence and have indicated that these are small-scale studies in specific contexts.

I have as suggested removed section 6 on the psychological factors involved in mobility and inserted the material elsewhere. I have added a discussion on broader road use and road safety issues in the section on ageing, health and mobility and attempted to respond to the the point that transport infrastrucure is a key enabling or disabling factor for mobility and social connection. 

I have also added the material from section 6 on the psychological factors involved in mobility to this section on health and mobility. I note the comment regarding chracterising consideration of risk as "fear" and have modified the language, although I do think that rational and emotional reactions are not always easy to disentangle. I have as suggested included the paragraph on security issues in the section on social support.

I have removed the reference to the Nigerian research as though I believe it could potentially form an interesting case study on the limitations of quantitaive research which is not based on prior qualitative inquiry, I do not feel that I have sufficient additional information to make this case clearly enough. 

Round 2

Reviewer 1 Report

Manuscript is still not conform to the PRISMA guidelines as well as the Manuscript Sections suggested by the journal:

Introduction, Materials and Methods, Results, Discussion

Author Response

I have rewritten the "materials and methods" section to respond to the points raised below. I have triued to conform more closely to PRISMA guidelines by including a breakdown of the material consulted and criteria used. I have retitled the manuscript sections as requested. 

Round 3

Reviewer 1 Report

This literature review is an important and an useful document for the understanding of the low and middle-income countries transport and mobility with impact to older adults.

Manuscript improved a lot since the first version.

Text needs minor text editing.